# Anogenital-Associated Papillomaviruses in Animals: Focusing on *Bos taurus* Papillomaviruses

**DOI:** 10.3390/pathogens9120993

**Published:** 2020-11-27

**Authors:** Nanako Yamashita-Kawanishi, Takeshi Haga

**Affiliations:** Division of Infection Control and Disease Prevention, Graduate School of Agricultural and Life Sciences, The University of Tokyo, Tokyo 113-0033, Japan; nanakoyamashita1906@g.ecc.u-tokyo.ac.jp

**Keywords:** anogenital papillomavirus, anogenital neoplasms, *Bos taurus* papillomavirus, bovine papillomavirus, anal fibropapilloma, vulval papilloma

## Abstract

In contrast to the diverse studies on human papillomaviruses (HPVs), information on animal PVs associated with anogenital lesions is limited. In the animal kingdom, papillomas occur more commonly in cattle than in any other animals, and diverse types of *Bos taurus* papillomaviruses (BPVs) exist, including the very recently discovered BPV type 29 (BPV29). From this perspective, we will review previous studies describing PV types associated with anogenitals in animals, with a focus on BPVs. To date, two classical BPV types, classified into *Deltapapillomavirus* (BPV1 and BPV2) and *Dyokappapapillomavirus* (BPV22), and two novel *Xipapillomaviruses* (BPV28 and BPV29) have been identified from anogenital lesions and tissues of the domestic cow. Due to the limited reports describing anogenital-associated PVs in animals, the relationships between their phylogenetic and pathogenetic properties are still undiscovered. Animal studies are valuable not only for the veterinary field but also for human medicine, as animal diseases have been shown to mimic human diseases. Studies of anogenital-associated PVs in animals have a positive impact on various research fields.

## 1. Introduction

Papillomavirus (PV), a double-stranded DNA virus, infects the basal layer of the epithelium, and its life cycle depends on the epithelial differentiation of the host tissue [1]. Infection with PVs has been associated with the development of cutaneous and/or mucosal neoplasms in various species, including humans and animals [2]. A PV genome comprises two late genes, several early genes, and a non-coding region known as the long control region (LCR) or the upstream regulatory region (URR) [3]. The classification of PVs is based on the nucleotide sequence similarity of the L1 open reading frame (ORF), the major capsid protein [2,4]. A novel PV genotype is considered if its L1 sequence shares less than 90% similarity with the closest known genotype [2]. Currently, in humans, more than 200 human PV (HPV) genotypes have been reported and are classified into five genera: *Alpha*-, *Beta-*, *Gamma-*, *Mu-*, and *Nu- papillomaviruses* [5,6,7]. Molecular epidemiological studies of HPV have revealed the genotypes accounting for the majority of cervical cancers and anogenital warts. HPV types 16 (57%) and 18 (16%) are the top two genotypes identified in cervical cancers and are the so-called high-risk HPVs [8,9]. Two HPV types, 6 and 11, known as low-risk HPVs, account for 90% of genital warts [10]. Although numerous discoveries of novel HPV types have been reported, epidemiological and pathological studies of HPVs enabled the clarification of the association between HPV type(s) and disease phenotype(s). These discoveries have led to the development of a prophylactic vaccine up to a nine-valent vaccine, Gardasil 9 (Merck & Co., Kenilworth, NJ, USA), showing protection against anogenital infections by nine inoculated HPV types [11]. 

In contrast to the diversified and progressing studies of HPVs, genomic features and the etiological significance of anogenital-associated PVs in animals are not well understood. Although studies describing the identification of anogenital-associated animal PVs in non-human primates [12,13,14], cetaceans [15,16,17,18,19,20,21], domestic animals [22,23,24,25,26,27,28,29,30], rodents [31,32,33,34], and bats [35] have been reported, the significant association between PV type(s) and disease phenotype(s) is still uncertain. Studies of animal PVs contribute not only to the veterinary field but also to human medicine, as animals are also regarded as animal models for human diseases. Due to the biological characteristics of species- and tissue specificity of PVs, it is challenging to study HPVs with laboratory animals. Following the discovery of cottontail rabbit PV (CRPV) [36] and rabbit oral PV (ROPV) [37] in the 1930s, researchers have evaluated the introduction of PV to its natural host (animal) to mimic the PV-induced anogenital neoplasms of humans [38]. In the animal kingdom, papillomas occur more commonly in cattle than any other animals, and diverse types of bovine (*Bos taurus*) PV (BPV) exist. Based on this background, we review anogenital-associated PVs in animals, focusing on BPVs, including our recent findings on novel BPV types.

## 2. Bovine Papillomaviruses and Anogenital Lesions

In cattle, tumors of the genitals have been reported since the 1950s worldwide [39,40,41,42,43,44,45,46,47]. Moreover, an outbreak of anal fibropapillomatosis following rectal palpation in a herd of beef heifers was announced in 1977 [48]. In 1980, BPV type 1 (BPV1) was detected from a bovine penile papilloma [26]. In a recent study, detection of BPV2 was confirmed in fibropapillomas collected from anal warts in two groups of heifers [24] (Table 1), suggesting the etiological relationship between BPV2 infection and anal fibropapilloma development. Although the examined samples did not derive from neoplastic lesions, BPV22, classified as the genus *Dyokappapapillomavirus*, was identified from vaginal biopsy/swab samples of a Holstein cow presenting vulvovaginitis [25] (Table 1). A recent paper also described a novel BPV from genital tract swabs of healthy dairy cows [49]. These studies have suggested the etiological roles of BPV in anogenital diseases of cattle.

Advancing technologies in the genomic research field have provided more opportunities to encounter novel PV types. Consensus primers are used to detect unknown PV types [51,52]. From anogenital tumors in cattle, we successfully detected two common *Deltapapillomaviruses* (BPV1 and BPV2) [23] and two unique BPVs classified in *Xipapillomavirus* [22] by using consensus primers targeting the L1 of BPVs [53,54]. *Deltapapillomavirus*-classified BPVs are suggested to be associated with fibropapilloma and *Xipapillomavirus*-classified BPVs, such as BPV3, BPV4, BPV6, and BPV12, have been shown to harbor epitheliotropic features [54,55,56].

### 2.1. Association of Bos taurus Papillomavirus Types 1 and 2 in Mucosal Sites

It has been demonstrated that BPV1 and BPV2, both classified as the species *Deltapapillomavirus* 4, have pathogenicity of fibropapilloma development in cattle [55]. The involvement of BPV1 and BPV2 in the development of cutaneous lesions has been reported [26,55], while their association with mucosal neoplasms has been limited. Of the mucosal sites, both BPV1 and BPV2 have been detected in urinary bladder tumors of cattle, which are proposed to be caused by immunosuppression via bracken fern grazing [57,58,59,60]. In our previous finding, exactly the same BPV1s were identified in both skin and some nodules in the upper gastrointestinal tract of a seven-month-old calf with severe papillomatosis, suggesting that BPV1 has the potential to develop lesions on both cutaneous and mucosal sites [61]. 

### 2.2. Association of Bos taurus Papillomavirus Types 1 and 2 in Anogenital Fibropapillomas

The first detection of BPV from anal lesions of heifers was reported from New Zealand [24]. In human studies, it is known that the distribution of the HPV variant differs between countries [62]. Therefore, it is essential to characterize PV sequences in wide geographic areas to clarify the association between disease phenotype(s) and PV genotype(s). In our previous study in Japan, BPV types 1 and 2 were identified in anal and vulval fibropapillomas, respectively, of Holstein Friesian dairy cows [23]. In this report, mRNA expression of the early genes (E2, E5, and E6), but not the late gene (L1) was determined in both BPV1 and BPV2 (BPV1/2)-identified cases. Our results suggested the presence of active virus, but without or with very little viral production [23]. Studies showing associations between BPV1/BPV2 and anogenital neoplasms extend the knowledge of BPV1/BPV2 tissue tropism, not only for the cutaneous sites but also for the mucosal sites. 

### 2.3. Identification of Two Novel Bos taurus Papillomavirus Types, 28 and 29, from Vulval Papillomas

Detection of novel BPVs has been reported worldwide. Our recent findings pertaining to genital tumor-associated BPV identified two novel *Xipapillomavirus*-classified BPV types, 28 and 29 [22]. Both types were identified in vulval papillomas of two Holstein Friesian dairy cows in Japan. 

BPV28 was identified from two vulval papilloma lesions adjacent to each other in one cow. An authentic BPV28 L1 sequence was identified in one sample, while BPV28 identified in the adjacent sample showed deletion of five nucleotides, resulting in two L1 ORFs [22]. L1 is known as the major capsid protein, comprising the viral capsid along with the L2 minor capsid protein. Previous studies showed that the N-terminal deletion of HPV16 L1 resulted in a mixed population of variable-sized capsids [63]. However, the significance of the BPV28 L1 frameshift mutation in this case remains unknown. In this cow, following the first sample collection, macroscopically similar lesions were observed and collected but no viral detection was confirmed [22]. 

Prior to this detection of BPV28, we characterized the complete sequence of BPV28, showing concurrent detection with BPV1 and BPV2, identified from a cutaneous papilloma excised from the face of a steer [64]. In humans, it is not unusual to see dual and/or multiple infections of high-risk and low-risk HPVs [65]. In our study, we could not evaluate the histopathological characteristics of BPV28-associated lesions due to the sample limitations. What we considered to be of interest was what the lesions of the concurrent detected cases consisting of fibropapilloma (*Deltapapillomavirus*)- and epitheliotropic (*Xipapillomavirus*)-associated BPVs looked like [64]. 

Another novel *Xipapillomavirus*-classified BPV type, 29, was identified from a vulval papilloma lesion from a dairy cow [22]. The L1 and complete nucleotide sequence of this novel BPV29 was the most identical to BPV6. The evolution of BPV29 remains undetermined, but it is interesting that the closest genotype, BPV6, has been identified in Japan, commonly detected in teat papillomatosis [54,66,67]. No recurrence of vulval papilloma was observed within the BPV29-identified cow [22]. 

In both BPV28- and BPV29-identified lesions, mRNA expression of the early genes (E1, E2, and E10) but not the late gene (L1) was observed [22], which was similar to our previous findings in anogenital fibropapilloma-associated BPV1 and BPV2 [23]. Moreover, immunohistochemical detection of BPV L1 antigen was negative in those anogenital cases, except for one vulval case showing PV antigen in a few differentiated keratinocytes. These results could be interpreted as showing that no, or very few, encapsulated viruses were being produced. In our study, since the available antibody for immunohistochemistry was produced against L1, detection against the early genes was not performed within the infected specimens. Previous studies have described the expression of BPV early proteins by immunohistochemistry on BPV-infected tissue sections [68,69] and by immunocytochemistry in cultured cells [68,70,71]. Additional analysis using antibodies targeting the early genes [68,69,70,71] would be helpful to further understand the pathogenicity of BPV in the naturally-occurring BPV-infected tissues. The presence of transcriptionally active papillomaviruses could become one of the key factors in viral transmissions [72]. Unlike the papilloma outbreaks reported previously [48,54], no anogenital papilloma lesions were observed within the analyzed herds in our study [22,23]. From the aspect of veterinary hygiene, it is important to keep in mind that breeding routines, such as rectal palpation, ultrasonography, the use of controlled internal drug release devices, and artificial insemination, could become a route of viral transmission. 

## 3. Papillomaviruses Identified from Anogenitals in the Other Animals

### 3.1. Non-Human Primates

In the early 1970s, preinvasive carcinoma cases of the non-human primate uterus were reported—an adenocarcinoma in situ of the endometrium in an aging chimpanzee (*Pan troglodytes*) and squamous cell carcinoma (SCC) in situ of the cervix in an aging crab-eating macaque (*Macaca fasicularis*) in the United States [73]. Dysplasia of the lower genital tract in the female macaques (*Macaca fasicularis*) from Southeast Asia was also reported [74]. However, in these two studies, no papillomaviral investigation was conducted. In 1990, rhesus papillomavirus 1 (RhPV1), currently designated as *Macaca mulatta* papillomavirus 1 (MmPV1) and classified as the species *Alphapapillomavirus* 12, was cloned and characterized from a lymph node metastasis of a primary penile carcinoma presented in a male rhesus monkey (*Macaca mulatta*) [13]. RhPV1 was identified in a group of rhesus monkeys showing sexual transmissions and viral integration in the tumor cell DNA [13], which has similar etiological characteristics to high-risk HPVs [75,76,77]. A serological study of RhPV1 was also conducted, but the relationship between the presence of RhPV1 antibody and cervical cancer in the animal remained uncertain [78]. In the late 1990s, twelve different types of RhPV were identified in DNA extracted from genital tissues of rhesus monkeys (*Macaca mulatta*) and long-tailed macaques (*Macaca fasciularis*), by PCR, diagnosed with cervical intraepithelial neoplasia (CIN) [12]. Later on, *Papio hamadryas* papillomavirus 1 (PhPV1) was identified from a CIN lesion from a hamadryas baboon (*Papio hamadryas anubis*) [14]. *Macaca fuscata* papillomavirus 2 (MfuPV2) was also characterized not only from (pre-) malignant lesions but also from the benign penile lesion of a Japanese macaque (*Macaca fuscata*) (GenBank accession number: MH469677) [6,50]. To our knowledge, PV genotypes associated with genital lesions of non-human primates are all classified as the genus *Alphapapillomavirus* (Table 1). 

### 3.2. Marine Animals

PVs identified from papillomas/wart lesions, normal mucosa, and collected fecal swabs of the anogenitals in marine animals have been reported (Table 1). Anogenital-associated PVs in animals are relatively rare, but various PVs—*Delphis delphis* papillomavirus 1 (DdPV1), *Phocoena phocoena* papillomaviruses (PphPVs) [15], *Phocoena spinpinnis* papillomavirus 1 (PsPV1) [16], and nine types of *Tursiops truncatus* papillomaviruses (TtPVs) [15,17,18,19,20,21]—have been identified. PVs derived from anogenital-associated samples of cetaceans are classified in either the genus *Upsilonpapillomavirus* (DdPV1, PphPV2, and TtPV1-4, TtPV7), *Omikronpapillomavirus* (PphPV1, PsPV1, TtPV5, TtPV6, and TtPV9), or *Dyopipapillomavirus* (PphPV4 and TtPV8) (Table 1). 

### 3.3. Horses

In horses, the etiological roles of *Equus caballus* papillomavirus 2 (EcPV2) are known and have been widely studied in equine genital neoplasms, including malignant tumors, such as SCCs [27]. Several other EcPV types, such as EcPV3, EcPV4, EcPV7, and EcPV9, have also been identified from equine genital-derived samples and lesions [28,29,30], but these genotypes do not seem to show metastatic behavior, unlike EcPV2 [79]. In HPVs, it is known that sexual transmission could become one of the risk factors for viral infection. In thoroughbred racehorse breeding, artificial breeding, such as by artificial insemination (AI) and embryo transfer (ET), is restricted by the criteria established by the International Stud Book Committee (ISBC). Therefore, it is important to consider the risk of EcPV transmission during natural mating of horses. 

### 3.4. Rodents 

From the aspect of human medical research, laboratory animals, including non-human primates and rodents, could become valuable surrogate models of human diseases. Considering the labor and costs of experiments involving non-human primates, the use of small rodents could become more practical. In rodents, two anogenital-associated papillomaviruses classified as the species *Pipapillomavirus* 2 have been identified (Table 1). *Mastomys coucha* papillomavirus 2 (McPV2) was the first genotype characterized from anogenital (anus, penis, or vulva) warts in rodents, but McPV2 was also detected from healthy tissues (liver, kidney, spleen, and intestine) of the studied animals [31]. 

A partial sequence of *Mus musculus* papillomavirus 1 (MmuPV1) (formerly MusPV) was identified in the tissues from florid and asymmetrical papillomas developed on the faces of nude mice (NMRI-*Foxn^nu^/Foxn1^nu^*) [32]. Later on, the complete genome of MmuPV1 was characterized and was shown to have the closest relationship to McPV2, with 80% L1 amino acid similarity [33]. The discovery of MmuPV1 extends to in vivo studies involving laboratory mice. MmuPV1 was first identified in cutaneous papilloma lesions but was also shown to infect mucosal tissue by delivering MmuPV1 into the vaginal tract or the anal canal of *Foxn^nu^/Foxn1^nu^* mice [34]. Moreover, MmuPV1 showed sexual transmission between unmanipulated, immunocompetent, male and female mice [80]. These studies demonstrate that MmuPV1 became a new model animal papillomavirus, contributing to the papillomaviral research field.

## 4. Vaccine Studies on BPVs

The development of the prophylactic HPV vaccine has led us to prevent HPV-associated cervical cancers and anogenital warts. The vaccines are composed of the self-assembled L1 major capsid protein of the target PV type expressed in eukaryotic cells, known as viral-like particles (VLPs) [81]. Prior to the successful development of HPV vaccines, as a surrogate model for humans, immunogenicity studies in animal PVs were demonstrated from the late 1930s [82]. From the 1950s, immunization studies have been demonstrated in bovids by injecting formalinized suspension of bovine wart materials into cattle [83,84,85]. Two classical BPV types, 1 and 2, have been studied for decades, showing potential to become a surrogate vaccine for humans and animals [81,86,87,88,89]. There is also published literature on immunogenicity studies on other BPV types besides BPV1 and BPV2. BPV4 has also been studied for vaccinology because its infection in cattle has been associated with papillomas of the alimentary canal mucosa, showing malignant progression [90] resembling that of human mucosal infections. Based on this etiological role of BPV4, previous studies have developed BPV4 vaccine candidates (recombinant BPV4 peptides and VLPs) and demonstrated protection against experimentally induced papillomas by immunizing calves [91,92]. BPV6 is recognized as the most frequently detected type of BPV in teat papillomatosis lesions [54,93]. From the viewpoint of the dairy industry, teat papillomatosis may be an obstacle to milking. The successful production of BPV6 VLPs and antigenicity in mice was confirmed in a previous study [94], demonstrating BPV6 VLPs as a vaccine candidate to protect from BPV6-associated teat/udder warts in cow. Although these animal studies contributed to the development of approved HPV vaccines, commercialized PV vaccines are not currently available for animals. For animals, especially cattle, cost-effective vaccines covering the wide-range of BPV types would be more practical to meet the demand of the vaccine. 

## 5. Conclusions

Molecular epidemiological studies and discoveries of anogenital-associated PVs in animals have been increasing in the 21st century. BPVs have been studied for decades, and the recent discoveries of BPVs, including the two classical types, 1 and 2, from anogenital neoplasms have suggested their potential to cause anogenital tumors in cattle. However, unknown aspects, including genomic characteristics of anogenital-associated PVs and disease-associated host factors, still remain in animals. Including our findings on two novel BPV types, exploratory studies of anogenital-associated lesions/samples may identify novel animal PV types. PV research related to large domestic animals, such as cattle and horses, will not only contribute to the PV research field but also to promoting better veterinary hygiene. We hope to see more studies increasing the knowledge on animal anogenital-associated PVs, leading to the enrichment of both veterinary and human medical fields.

## Figures and Tables

**Table 1 pathogens-09-00993-t001:** Classification and source of papillomaviruses identified in anogenital-related samples in animals.

Host Species (Taxonomic Order)	Host Species (Common Name)	Papillomavirus Type	Abbreviation (Formerly)	Classification	Source	Reference
*Artiodactyla*	*Bos taurus* (Domestic cow)	*Bos taurus* papillomavirus 1	BPV1	*Deltapapillomavirus* 4	Anal fibropapilloma	[23]
					Penile papilloma	[26]
		*Bos taurus* papillomavirus 2	BPV2	*Deltapapillomavirus* 4	Anal fibropapilloma	[24]
					Vulval fibropapilloma	[23]
		*Bos taurus* papillomavirus 22	BPV22	*Dyokappapapillomavirus*	Vulvovaginitis, vaginal swabs	[25]
		*Bos taurus* papillomavirus 28	BPV28	*Xipapillomavirus* 1	Vulval papilloma	[22]
		*Bos taurus* papillomavirus 29	BPV29	*Xipapillomavirus* 1	Vulval papilloma	[22]
*Cesatea*	*Delphis delphis* (Short-beaked common dolphin)	*Delphis delphis* papillomavirus 1	DdPV1	*Upsilonpapillomavirus* 1	Penile wart	[15]
	*Phocoena phocoena* (Harbor porposie)	*Phocoena phocoena* papillomavirus 1	PphPV1	*Omikronpapillomavirus* 1	Penile papilloma	[15]
		*Phocoena phocoena* papillomavirus 2	PphPV2	*Upsilonpapillomavirus* 1	Penile papilloma	[15]
		*Phocoena phocoena* papillomavirus 4	PphPV4	*Dyopipapillomavirus* 1	Penile papilloma	[15]
	*Phocoena spinpinnis* (Burmeister‘s porposie)	*Phoconea spinpinnis* papillomavirus 1	PsPV1	*Omikronpapillomavirus* 1	Genital papilloma	[16]
	*Tursiops truncatus* (Bottlenose dolphin)	*Tursiops truncatus* papillomavirus 1	TtPV1	*Upsilonpapillomavirus* 1	Penile wart	[17]
		*Tursiops truncatus* papillomavirus 2	TtPV2	*Upsilonpapillomavirus* 2	Genital lesion	[18]
		*Tursiops truncatus* papillomavirus 3	TtPV3	*Upsilonpapillomavirus* 1	Penile wart	[17]
		*Tursiops truncatus* papillomavirus 4	TtPV4	*Upsilonpapillomavirus* 1	Genital lesion	[19]
		*Tursiops truncatus* papillomavirus 5	TtPV5	*Omikronpapillomavirus* 1	Genital lesion	[19]
		*Tursiops truncatus* papillomavirus 6	TtPV6	*Omikronpapillomavirus* 1	Genital lesion	[19]
		*Tursiops truncatus* papillomavirus 7	TtPV7	*Upsilonpapillomavirus* 1	Normal genital mucosa	[19]
		*Tursiops truncatus* papillomavirus 8	TtPV8	*Dyopipapillomavirus* 1	Fecal swab	[20]
					Penile papilloma (partial L1 sequence)	[15]
		*Tursiops truncatus* papillomavirus 9	TtPV9	*Omikronpapillomavirus* 1	Fecal swab	[21]
*Chiroptera*	*Minoiopterus schreibersii* (Common bent-wing bat)	*Minoiopterus schreibersii* papillomavirus 1	MscPV1	Unclassified	Anal swab or pharyngeal swab	[35]
	*Myotis riketti* (Rickett’s big-footed bat)	*Myotis riketti* papillomavirus 1	MrPV1	Unclassified	Anal swab or pharyngeal swab	[35]
*Perissodactyla*	*Equus ferus caballus* (Domestic horse)	*Equus caballus* papillomavirus 2	EcPV2	*Dyoiotapapillomavirus* 1	Genital neoplasia, squamous cell carcinoma	[27]
		*Equus caballus* papillomavirus 3	EcPV3	*Dyorhopapillomavirus* 1	Penile/preputial papilloma	[28]
		*Equus caballus* papillomavirus 4	EcPV4	*Dyoiotapapillomavirus* 2	Vulval and inguinal plaques	[29]
		*Equus caballus* papillomavirus 7	EcPV7	*Dyoiotapapillomavirus* 1	Penile mass	[29]
		*Equus caballus* papillomavirus 9	EcPV9	Unclassified	Semen (stallion with a penile wart)	[30]
*Primates*	*Macaca fascicularis* (Cynomolgus macaque)	*Macaca fascicularis* papillomavirus 3	MfPV3 (RhPV-d)	*Alphapapillomavirus* 12	Cervical intraepithelial neoplasia	[12]
			MfPV4	*Alphapapillomavirus* 12	Cervical intraepithelial neoplasia	[12]
			MfPV5 (MfPV-a)	*Alphapapillomavirus* 12	Cervical intraepithelial neoplasia	[12]
			MfPV8 (RhPV-a)	*Alphapapillomavirus* 12	Cervical intraepithelial neoplasia	[12]
	*Macaca fuscata* (Japanese macaque)	*Macaca fuscata* papillomavirus 2	MfuPV2	*Alphapapillomavirus*	Benign penile lesion	[50] (Unpublished)
	*Macaca mulatta* (Rhesus macaque)	*Macaca mulatta* papillomavirus 1	MmPV1 (RhPV1)	*Alphapapillomavirus* 12	Mucosal genital carcinoma	[13]
	*Papio hamadaryas anubis* (Hamadryas baboon)	*Papio hamadaryas* papillomavirus 1	PhPV1	*Alphapapillomavirus* 12	Cervical intraepithelial neoplasia	[14]
*Rodentia*	*Mastomys coucha* (Southern multimammate mouse)	*Mastomys coucha* papillomavirus 2	McPV2	*Pipapillomavirus* 2	Anogenital wart and healthy tissues	[31]
	*Mus musculus* (*Foxn^nu^/Foxn1^nu^* nude mice)	*Mus musculus* papillomavirus 1	MmuPV1	*Pipapillomavirus* 2	Vaginal, cervical, anal dysplasia (Experimental infection of MmuPV1)	[34]

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
