# Peer review of "Anogenital-Associated Papillomaviruses in Animals: Focusing on Bos taurus Papillomaviruses"

_pathogens, 2020, doi:10.3390/pathogens9120993_

Round 1

Reviewer 1 Report

Although human papillomaviruses have been intensely studied due to their association with cancer, animal papillomaviruses have had less attention. In this interesting review, Yamashita-Kawanishi et al discuss genital papillomavirus infections of animals. The content is interesting and it is generally well presented. A few revisions will make it more suitable for publication:

  1. The paper is generally quite readable, but there are numerous small grammatical and wording errors that are distracting. I recommend that a native English speaker read it over carefully, preferably someone with technical knowledge.
  2. Abbreviations should be defined the first time that they are used. There are several examples of abbreviations that are not defined, or defined after the first use. The use of BPV in the abstract is one example.
  3. Line 61 - please indicate that the study referenced in reference 48 was in 1977. Also, would the authors have expected PCR examination of papillomaviruses in those early studies?
  4. “Cattle” is plural. “Cow” is singular for a female which has had one calf and “heifer” is a female who has not had a calf. “Bull” is singular for a male, and “steer” is a castrated male. These distinctions cause some confusion. For example, Line 120 “no recurrence…BPV29-identified cattle” does that mean a group of cattle or do the authors mean an individual cow?
  5. Genus names are capitalized and species names are not, and both are italicized. For example line 141 Macaca mulatta.
  6. Line 155 why would one do a fecal swab of genitals?
  7. Lines 205-205 – E7 disrupts function of Rb and degrades the protein, but “degrading its function” is odd.

Author Response

Thank you very much for your valuable comments on our manuscript.

Please see the attached file for the details. Best regards, Takeshi Haga

Reviewer 2 Report

ASSOCIATION OF BOVINE PAPILLOMAVIRUSES IN ANOGENITAL LESIONS OF CATTLE

Summary

PVs’ infection has been associated with the development of cutaneous/mucosal neoplasms in various species, both humans and animals. Animal PVs are widely studied also as surrogate models of human disease, although the knowledge in animal anogenital-associated PVs needs to be improved. In details, the significant association between PV type and disease phenotype in anogenital-associated animal PVs is still uncertain. In the animal kingdom, cattle more frequently have PV’s induced lesions, and furthermore the PVs species associated to anogenital lesions are classified in the genera Deltapapillomavirus and Xipapillomavirus. BPV1 and BPV2 (Deltapapillomavirus) were detected in anogenital fibropapillomas, whereas BPV28 and BPV29 (Xipapillomavirus) in vulval papillomas. In both cases, mRNA expression of the early genes but not for the late gene was detected, suggesting the presence of active virus, but without or with very few viral production. Further investigation about PVs’ induced anogenital lesions in cattle are certainly needed, also for veterinary hygiene related risks (PVs’ infections via fomites).

In non-human primates several species of anogenital PVs were identified, all of them belonging to the Alphapapillomavirus genus, inducing not only malignant/pre-malignant lesions (RhPV1 or MmPV1, PhPV1) but also benign penile lesion (MfuPV2).

In marine mammals the PVs-associated anogenital lesions are quite rare, even though PVs from the genera Upsilonpapillomavirus (DdPV1, PphPV2, TtPV1-4, TtPV7), Omikronpapillomavirus (PphPV1, PsPV1, TtPV5, TtPV6, TtPV9) and Dyopipapillomavirus (PphPV4 and TtPV8) are reported.

In horses, genital lesions/neoplasms are frequently induced by EcPV2 (Dyoiotapapillomavirus), but several others PVs have been identified from genital-derived lesions (EcPV3, EcPV4, EcPV7, and EcPV9).

In rodents, two anogenital-associated papillomaviruses classified in the Pipapillomavirus genus have been identified: McPV2 and MmuPV1. MmuPV1 infects both the dermis and the mucosal tissue, and showed sexual transmission between immunocompetent mice. For these reasons, MmuPV1 is a new suitable animal model, contributing to papillomaviral research field.

Comments

An interesting review, which analyses the association between Papillomaviruses and anogenital lesions in cattle and other mammals, highlighting the importance of animal models for the human papillomaviral disease knowledge. Nevertheless, there are several grammar errors and misprint all over the text, and so minor revisions are required (For instance lines 14, 72, 105, 110, 134 and 356). Furthermore, some references might be improved (For instance line 73: there is no reference to Xipapillomavirus in the prior part of the text).

Author Response

(The authors gave the same response as above.)

Reviewer 3 Report

Yamashita- Kawanishi and Haga review bovine papillomaviruses and their involvement in genital lesions.  These authors previously isolated and sequenced several BPV types.

  1. The grammar needs improvement throughout the manuscript. Some sentences were difficult to comprehend.

  1. The authors might consider a broader title as the article describes several other non-human animal PVs in relationship to genital infections in Section 3m which amounts to about half of the article.

  1. Lines 36-39 seem superfluous and too detailed. Why would a reader care about HPV52, for example?

  1. The authors mention the HPV vaccine but not the BPV vaccine literature. There are many reports dating back several decades that are still relevant.

  1. The authors refer to their work in which early gene transcripts were identified (refs 22 and 23). There are published antibodies to several BPV-1 early proteins. Have these ever been tested with infected specimens using immunohistochemistry by the authors or any other groups? In that regard, it would seem relevant to compare the early proteins for conservation among the various BPV genotypes.

  1. Lines 201-215 discuss the E7 motif LXCXE, which is required for binding to the Rb protein. That seems like a digression from the main topic of the paper. However, if that were the authors’ intent, it would be worthwhile in this context to also compare the other early proteins E6, E1 and E2, and whether there is an E5, among the BPV genotypes. A Table or Figure depicting these protein sequences and common motifs would be informative.  For example, do any of these have homology to the region of E6 that binds to E6AP, p53, or MamL1?

Author Response

(The authors gave the same response as above.)

Round 2

Reviewer 3 Report

The authors have improved the manuscript. The grammar still needs to be fixed for example line 12-13 or line 114 (considerate) and there are more.

There is old literature on vaccination - look up Carl Olson and bovine PV in PubMed that the authors should consider. 

The response to early protein detection in BPV was inadequate. I understood what the L1 antibody is.  Have they or others used existing antibodies to BPV early proteins E6, E7, E1, E2 or E5 by immunocytochemistry on tissues or in raft cultures. I believe those have been published. The authors might state that these could be used to study BPV infected tissues from warts as a model to understand early PV protein expression.
